# Turning the tide: a quasi-experimental study on a coaching intervention to reduce burn-out symptoms and foster personal resources among medical residents and specialists in the Netherlands

Lara Solms ,[1,2] Annelies van Vianen,[1] Jessie Koen,[1] Tim Theeboom,[3] Anne P J de Pagter,[4,5] Matthijs De Hoog,[6] On behalf of the Challenge & Support Research Network

APJdP and MDH contributed equally.

This research has been presented at the following international conferences: Coaching in Leadership and Healthcare Conference, Institute of Coaching, Boston, USA (29 September 2018). International Association for Medical Education AMEE Conference, Vienna, Austria (26 August 2019).

For numbered affiliations see end of article.

**Correspondence to**
Dr Anne P J de Pagter;
p.depagter@erasmusmc.nl

## ABSTRACT

**Objectives** Physician burn-out is increasing, starting already among residents. The consequences of burn-out are not limited to physicians' well-being, they also pose a threat to patient care and safety. This study investigated the effectiveness of a professional coaching intervention to reduce burn-out symptoms and foster personal resources in residents and specialists.

**Design** In a controlled field experiment, medical residents and specialists received six coaching sessions, while a control group did not undergo any treatment. The authors assessed burn-out symptoms of exhaustion and cynicism, the personal resources psychological capital, psychological flexibility and self-compassion, as well as job demands and job resources with validated questionnaires (January 2017 until August 2018). The authors conducted repeated measures analyses of variance procedures to examine changes over time for the intervention and the control group.

**Setting** Four academic hospitals in the Netherlands.

**Participants** A final sample of 57 residents and specialists volunteered in an individual coaching programme. A control group of 57 physicians did not undergo any treatment.

**Intervention** Coaching was provided by professional coaches during a period of approximately 10 months aiming at personal development and growth.

**Results** The coaching group (response rate 68%, 57 physicians, 47 women) reported a reduction in burn-out symptoms and an increase in personal resources after the coaching intervention, while no such changes occurred in the control group (response rate 35%, 42 women), as indicated by significant time × group interactions, all p<0.01. Specifically, physicians increased their psychological capital ($\eta_p^2$=0.139), their self-compassion ($\eta_p^2$=0.083), and reported significantly less exhaustion ($\eta_p^2$=0.126), the main component of the burn-out syndrome.

**Conclusion** This study suggests that individual coaching is a promising route to reduce burn-out symptoms in

### Strengths and limitations of this study

► This study provides first evidence from a controlled intervention study on the effectiveness of coaching in both medical residents and specialists.

► Six individual professional face-to-face coaching sessions can decrease burn-out symptoms (ie, exhaustion) among medical residents and specialists.

► Preventive coaching contributes to the improvement of the personal resources psychological capital and self-compassion, resources that play a role in the prevention of burn-out.

► The study is limited by its quasi-experimental design. However, the analyses controlled for initial differences between the coaching and the control group.

► The coaching group consisted exclusively of paediatric residents and physicians. Consequently, more research is needed that evaluates the effectiveness of coaching in different specialties, allowing broader generalisation for coaching effectiveness among healthcare professionals.

both residents and specialists. Moreover, it strengthens personal resources that play a crucial role in the prevention of burn-out.

## INTRODUCTION

Physicians experience a variety of stressors including time pressure, emotionally taxing patient interactions and an increasing bureaucratic burden. Not surprisingly, burn-out (ie, feeling exhausted, dissociated and less efficient) is high among senior healthcare professionals as well as residents.[1 2] Burn-out has severe consequences for physicians, often leading to long-term absenteeism and

eventually abandonment of the medical profession.[3] But the negative consequences are not limited to physicians' well-being and careers: with burn-out flooding the healthcare system, patient safety is also at risk. Physician burn-out is associated with poorer quality of care and reduced patient safety.[4 5]

In order to reduce the risk of physician burn-out, and thus warrant adequate patient care and patient safety, powerful interventions are needed that prioritise physicians' needs. This is the case in professional coaching, which is commonly defined as 'a result-oriented, systematic process in which the coach facilitates the enhancement of life experience and goal attainment in the personal and/or professional life of normal, non-clinical clients.'[6] This definition of coaching acts on the assumption that coaching is a facilitative process aimed a self-directed change of the client.[7] Additionally, this definition distinguishes coaching from other helping relationships such as mentoring and counselling.[8] Mentoring generally refers to a relationship between a more senior employee and a protégé aimed at offering guidance and feedback in a specific organisational context.[9] In coaching, a coach usually does not hold a formal position within the client's organisation. Additionally, our definition of coaching emphasises a non-clinical target group, which makes it clearly distinguishable from counselling and therapy.

Surprisingly, coaching is not common in medical practice and research is scarce[10–14] despite the fact that the positive effects of coaching on well-being and functioning have been demonstrated in a number of educational and professional settings.[15] Furthermore, with coaching being generally connected to problem elimination (eg, burn-out) in healthcare, rather than to professional development and well-being, its power is underrated if not invisible due to stigma. Given the potential benefits of coaching for physician well-being, research on the effectiveness of coaching in a professional development setting is sorely needed.

A professional coaching intervention may simultaneously help to resolve and prevent burn-out among physicians. That is, professional coaching can not only directly reduce burn-out symptoms, but can also strengthen personal resources that may prevent such burn-out symptoms in the first place.[16] This assumption is rooted in research on burn-out, which shows that the onset of burn-out is caused by both heavy job demands and a lack of (personal) resources.[17] Personal resources refer to 'aspects of the self that are generally linked to resiliency and refer to individuals' sense of their ability to control and impact on their environment successfully'.[18] According to the Job Demands-Resources Model (JD-R),[17] a common work-stress model in the prediction of burn-out and work engagement, personal resources help people to deal with extreme demands, ultimately buffering the negative effects of job demands on burn-out.[19] At the same time, personal resources stimulate motivation and work engagement. With both work engagement and well-being (ie, a lack of burn-out) being indispensable for

optimal physician functioning, the value of professional coaching lies in its ability to kill two birds with one stone: It aims to reduce burn-out symptoms as well as stimulate lifelong reflection and self-management through recognising and strengthening individuals' personal resources.

In this two-wave quasi-experimental study, we evaluated the benefits of an individual coaching programme for the resources, demands and well-being (ie, lack of burn-out symptoms), and work engagement of medical residents and specialists in the Netherlands.

## METHOD

### Study setting and population

This study evaluates the effectiveness of an individual coaching programme in two major academic hospitals, the Erasmus Medical Center (EMC) and the Leiden University Medical Center (LUMC) in the Netherlands. Using an quasi-experimental pretest and posttest control design, this study comprises the comparison of a treatment group (ie, coaching group) with a control group that did not receive any treatment on two measurement occasions (ie, at pretest and posttest). In a quasi-experimental design like this, the assignment to conditions (ie, coaching vs no coaching) is non-random.[20] A final number of 114 physicians participated in this study of which 57 received individual coaching between January 2017 and August 2018. The coaching programme was completely voluntary, offering six individual coaching sessions to both residents and specialists from the paediatrics department at the EMC and LUMC. Because funding for the coaching programme was initially only available for the paediatrics department, physicians from other departments (ie, internal medicine, neurology) and paediatric residents from two other hospitals (ie, VU University Medical Center (VUmc) and Academic Medical Center Amsterdam (AMC)) served as a control group. Additionally, paediatricians who did not voice interest in the coaching programme were placed in the control group as well. See table 1 for sample characteristics.

### Intervention and procedure

Physicians were informed through different channels (ie, email newsletter, information presentation, mouth to mouth) about the coaching programme and could sign themselves up for the programme via email. Physicians that voiced interest in the coaching programme were asked to participate in the study and were able to choose a coach of their preference. All coaches participating in the programme were selected based on a number of relevant criteria such as years of experience and affinity and experience with the medical profession. Specifically, all coaches were selected based on their senior level of coaching experience, their experience with physician-clients, positive references from previous physician clients, and accredited coaching training. The selection committee consisted of a coaching professional, a senior human resources manager, and the medical specialist

**Table 1** Characteristics of the study population in a study on coaching effectiveness for medical residents and specialists, 2017-2018*

| Characteristics | Intervention (N=57) | Control (N=57) |
|---|---|---|
| Male sex-no (%) | 10 (17.5) | 15 (26.3) |
| Age-year | | |
| Median | 33 | 35 |
| IQR | 9.5 | 12 |
| Specialty-no (%) | | |
| Paediatrics | 57 (100) | 32 (56.1) |
| Internal medicine | – | 15 (26.3) |
| Neurology | – | 10 (17.5) |
| Professional role-no (%) | | |
| Resident | 33 (57.9) | 36 (63.2) |
| Specialist | 24 (42.1) | 21 (36.8) |
| Hospital-no (%) | | |
| EMC | 32 (56.1) | 33 (57.9) |
| LUMC | 25 (43.9) | 9 (15.8) |
| VUmc | – | 7 (12.3) |
| AMC | – | 8 (14.0) |
| Coaching experience-no (%) | 22 (38.6) | 19 (33.3) |
| Home situation-no (%) | | |
| Children, one or more | 28 (49.1) | 29 (50.9) |
| No children | 29 (50.9) | 28 (49.1) |

*This study was conducted at four academic hospitals in the Netherlands. In this study, the authors investigated the effects of an individual coaching intervention on burn-out symptoms, work engagement, personal resources, job demands and job resources among paediatric residents and specialists.
AMC, Academic Medical Center Amsterdam; EMC, Erasmus Medical Center; LUMC, Leiden University Medical Center; VUmc, VU University Medical Center.

and initiator of the coaching programme. Physicians could view introductory videoclips of coaches on the programme website. In these 1 min long videos, coaches introduced themselves and provided information about their way of working with clients. Thereafter, physicians chose their coach and the first coaching session was arranged.

### The coaching process

Coaches and participants received ample freedom to shape the coaching programme according to coaches' professional methods and participants' needs. Because an important premise of successful coaching is that the coach and the client agree on the goals to achieve, as well as the means to achieve them,[21 22] we largely avoided regulations to the coaching process (such as the topics of the coaching, the coaching method or the speed of the trajectories) that might have stood in the way of such consensus. Constraints were set only with regard to the

overall outline of the coaching programme. That is, coaching was set to a maximum of 6 (1 or 1.5 hours long) sessions and coaches and participants were encouraged to complete the coaching trajectories within a period of approximately 10 months but could stretch their trajectories if necessary (M=7.98, SD=2.81), which only few participants did. All participants started their coaching trajectory individually depending on the availability of their coach. Time in between coaching sessions was determined by the participants—and hence varied—and was further not registered. All coaching sessions took place face-to-face and outside of work at the coach's workspace. Informed consent was obtained from all participants in both the coaching and the control group at the beginning of the study. Participants who did not give consent, were excluded from the study. Demographics as well as the study variables were measured with an online survey delivered via Qualtrics (Qualtrics, 2005) shortly before the first coaching session at baseline (T1) and minimal 7 days (M=87.25, SD=92.95, range: 7–364) after the last coaching session was finished (T2). Participants who failed to fill out the T1 or T2 survey at first, received up to three reminders by email with the request to complete the survey. For a description of exclusion criteria, see figure 1.

### Study variables

In line with the JD-R model, we measured job demands (workload, job insecurity, work–family conflict), job resources (autonomy, colleague support, supervisor support), personal resources (psychological capital, self-compassion, psychological flexibility), as well as burn-out symptoms and work engagement.

#### Job demands

We measured workload, job insecurity, and work-family conflict.

Workload was assessed with four items from the Quantitative Workload Inventory[23] and two additional items that were added to match the specific demands of medical practice. The two additional items assessed working overtime and emotional strain. All items were measured on a 5-point scale ranging from 1 ('never') to 5 ('always').

Job insecurity, that is, 'the perceived threat of job loss and the worries related to that threat' was assessed with a 5-item adapted version of the Job Insecurity Scale.[24] The items were scored on a 7-point scale ranging from 1 ('not at all applicable') to 7 ('very applicable'). Work–family conflict was measured with four items of the Work–Family Conflict Scale[25] assessing 'the general demands of, time devoted to, and strain created by the work interfere with performing family-related responsibilities.' The items were scored on a 7-point scale ranging from 1 ('not at all applicable') to 7 ('very applicable').

#### Job resources

Job resources encompassed autonomy, supervisor support and colleague support.

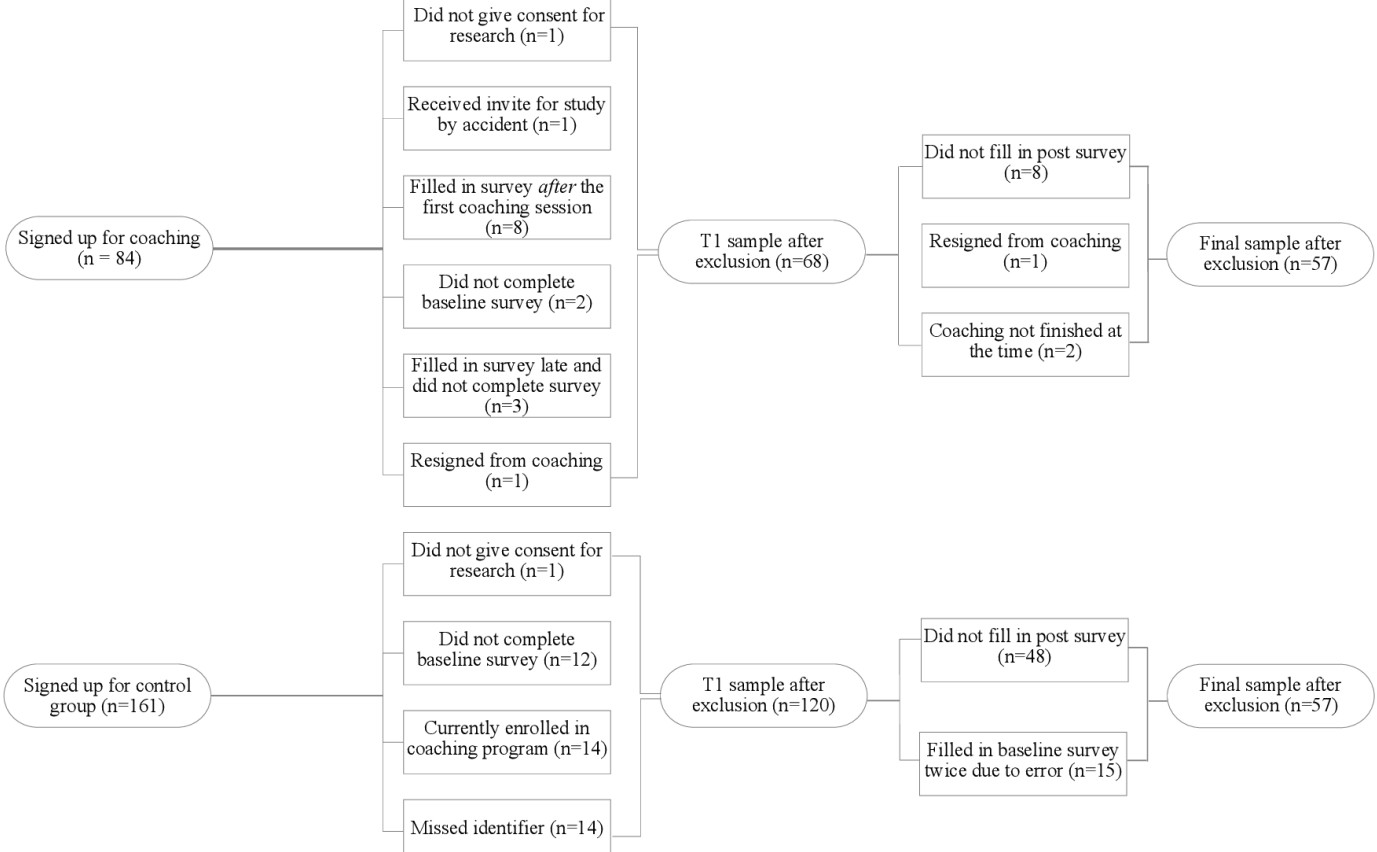

**Figure 1** Flow chart of study inclusion for participants in coaching and control group in a study on coaching effectiveness for medical residents and specialists.

Autonomy was measured with nine items from the Work Design Questionnaire[26] assessing perceived autonomy with regard to work scheduling and methods, and decision making. The items were scored on a 7-point scale ranging from 1 ('totally disagree') to 7 ('totally agree').

Supervisor support, that is, the experienced psychological and work support from the supervisor was assessed with six items from Vinokur, Schul and Caplan.[27] For residents, supervisory support measured the support received from the training supervisor, whereas for specialists, supervisory support measured the support received from the head of the department. The items were scored on a 7-point scale ranging from 1 ('totally disagree') to 7 ('totally agree').

Colleague support, the experienced psychological and work support from colleagues, was assessed with the same six items as supervisor support,[27] but the items referred to colleagues instead of the supervisor.

### Personal resources
We measured psychological capital, self-compassion and psychological flexibility.

To capture psychological capital's components, hope, optimism and resilience, we used nine items from the Dutch version of the PsyCap questionnaire.[28] To measure the fourth component, self-efficacy, we used three items based on the Generalised Self-Efficacy Scale[29] that were adapted so they would fit the occupational setting as used in previous research.[30] The items were scored on a 7-point scale ranging from 1 ('totally disagree') to 7 ('totally agree').

Self-compassion, that is 'treating oneself with kindness, recognising one's shared humanity and being mindful when considering negative aspects of oneself' was measured with six items from the Self-Compassion Scale.[31] This scale encompasses three subscales: self-kindness, common humanity and mindfulness. The items were scored on a 5-point scale ranging from 1 ('rarely') to 5 ('almost always').

Psychological flexibility, that is, the ability to flexibly take appropriate action towards achieving goals and values, even in the presence of challenging or unwanted events was measured with seven items of the Work Acceptance and Action Questionnaire.[32] The items were scored on a 5-point scale ranging from 1 ('rarely') to 5 ('almost always').

### Burn-out symptoms and work engagement
We measured burn-out symptoms with the two core scales exhaustion and cynicism of the Dutch version of the Maslach Burnout Inventory.[33 34] Both scales were measured with five and four items, respectively. The items were scored on a 7-point scale ranging from 1 ('totally disagree') to 7 ('totally agree').

We measured work engagement with the Utrecht Work Engagement Scale.[35] Its nine items cover the three subscales vigour, dedication and absorption. The items were scored on a 7-point scale ranging from 1 ('never') to 7 ('always').

## Statistical analyses
### Intervention effects
To test if the coaching intervention would have beneficial effects, repeated measures analyses of variance (ANOVAs) procedures were performed to examine changes over time for the intervention and the control group. The outcomes analysed were job demands (workload, job insecurity, work–family conflict), job resources (autonomy, colleague support, supervisor support), personal resources (psychological capital self-compassion, psychological flexibility), as well as burn-out symptoms (exhaustion, cynicism) and work engagement. We controlled for coaching attitude, that is, the degree to which one believes coaching is beneficial or helpful, which was measured at baseline, because it can be expected that a positive attitude may contribute to the success of the intervention. Significant Time x Group interactions of the outcome variables were followed up with post hoc tests.

### Preliminary analyses
Self-selection of participants: Because participation in the coaching programme was voluntarily—and complete randomisation of participants to conditions was not possible due to internal (ie, financial and time) restrictions and prior agreements within the hospital organisations—we examined structural demographic differences prior to the intervention between the coaching and the control group (T1). These demographics were gender, age, tenure (ie, medical resident, specialist), department (ie, paediatrics, internal medicine, neurology) and hospital affiliation (ie, EMC, LUMC, VUmc, AMC). Sample characteristics are displayed in table 1. While both groups did not differ with respect to gender ($x^2(1)$=1.28, p=0.26), age (F (1,112)=0.49, p=0.49) and tenure ($x^2(1)$=0.33, p=0.57), they did differ in department affiliation ($x^2(2)$=32.02, p<0.001) and hospital affiliation ($x^2(3)$=22.55, p<0.001). More specifically, all coaching participants were affiliated with the paediatrics department of two of the four participating hospitals. We conducted three types of additional analyses to rule out that potential effects attributed to the coaching intervention were caused by factors related to the imbalance of department and hospital affiliation—although conceptually, this is highly unlikely.

Hospital affiliation; To estimate a potential impact of hospital affiliation on treatment effectiveness, we conducted multiple univariate repeated measures for each of the outcome variables including hospital affiliation as additional control variable to see if the previous results would hold. Additionally, we conducted the original analyses solely for physicians employed at the two medical hospitals that were represented in the intervention group.

**Table 2** Means and SD of study variables for the intervention and the control group at baseline (T1) in a study on coaching effectiveness for medical residents and specialists, 2017–2018*

| Study variables | Intervention (n=57) Mean (SD) | Control (n=57) Mean (SD) |
|---|---|---|
| Job demands | | |
| Workload** | 3.48 (0.67) | 3.10 (0.78) |
| Job insecurity** | 4.24 (1.33) | 3.37 (1.45) |
| Work–family conflict** | 4.85 (1.05) | 4.00 (1.19) |
| Job resources | | |
| Autonomy | 4.39 (1.03) | 4.67 (1.14) |
| Colleague support | 5.33 (0.96) | 5.47 (0.90) |
| Supervisor support | 4.63 (1.51) | 4.98 (1.42) |
| Personal resources | | |
| PsyCap** | 4.83 (0.69) | 5.19 (0.72) |
| Self-compassion** | 3.07 (0.60) | 3.39 (0.66) |
| Psych. flexibility* | 3.43 (0.63) | 3.67 (0.53) |
| Outcomes | | |
| Exhaustion** | 2.75 (1.08) | 2.13 (0.92) |
| Cynicism | 2.11 (1.08) | 2.06 (0.93) |
| Work engagement | 5.08 (0.78) | 5.04 (0.75) |

Differences in means between the intervention and the control group are indicated by the following significance values: *p<0.05; **p<0.01.
PsyCap, psychological capital.

Department affiliation; Given that all participants in the coaching intervention were affiliated with the paediatrics department we analysed whether paediatricians differed from physicians affiliated with other departments (eg, neurology, internal medicine) with respect to contextual variables, here competition, and psychological safety, variables that reflect experienced department work climate and potentially could influence treatment effectiveness.

### Baseline differences between groups
With respect to the outcome variables at baseline, we found significant differences between the intervention and the control group: The intervention group scored significantly lower on personal resources, and significantly higher on job demands and exhaustion, similar to the results of a previous study on counselling in Norwegian doctors.[36] An overview of the differences between the groups is displayed in table 2. Because distribution of participants was not random, and because there were significant differences on a number of outcomes prior to the intervention, we tested our hypotheses with repeated measures ANOVA. These analyses are favoured over the ANOVA in a non-randomised intervention study.[37] Additionally, we followed the recommendations of Huberty and Moris[38] and conducted multiple ANOVAs as opposed

to a multivariate ANOVA as a preliminary step to multiple ANOVAs.

## Patient and public involvement

This study investigated the effectiveness of a professional coaching intervention in medical residents and specialists. No patients or public representatives were involved in the study.

## RESULTS

A total number of 84 physicians signed up for the coaching programme while 161 physicians signed up for the control group. Of these two groups, 57 physicians in each group completed the follow-up measurement and were included in the final sample (figure 1). Table 1 shows the demographic characteristics of the study population. Internal consistencies ranged from 0.72 to 0.95 and were acceptable for all scales. See table 3 for correlations between the study variables at baseline.

### Intervention effects

The analyses revealed significant changes in the intervention group that did not occur in the control group, as indicated by significant group × time interactions for a number of outcomes. A summary of the results is shown in table 4. With regard to job demands, post hoc analyses revealed a decrease in job insecurity and work-family conflict in the intervention group with both p<0.05. With regard to job resources, post hoc analyses showed that autonomy increased in the intervention group, while supervisor support decreased in the control group, all p<0.05. With regard to personal resources, post hoc comparisons indicated an increase in psychological capital and self-compassion in the intervention group, all p<0.05, as well as a decrease in self-compassion in the control group, p<0.05. No changes occurred in psychological flexibility, in either the control or coaching group, all p>0.05. Finally, with regard to outcomes, analyses showed that the coaching group significantly decreased their burn-out symptoms but showed no changes in work engagement. Post hoc comparisons indicated a decrease in exhaustion in the intervention group, p<0.05, while no such changes occurred in the control group, all p>0.05 or with regard to cynicism, p>0.05. For a graphical representation of these effects, see figure 2.

### Supplementary analyses

In order to rule out that effects attributed to the intervention were (partly) influenced by hospital and department affiliation we conducted three additional analyses. (Tables summarising the results can be requested from the first author)

Hospital affiliation; First, we conducted repeated measures analyses for each outcome variable with the whole sample, but this time added hospital affiliation as a control variable. The results of these analyses revealed no significant differences with those of the original analyses,

except for work engagement as outcome. Here, we find (instead of a marginal significant) a significant group × time interaction, p=0.026. Post hoc analyses indicated that the coaching group reported increased work engagement after the coaching programme, with no changes occurring in the control group. Overall, these results indicate that hospital affiliation did not influence treatment effectiveness in significant ways. Additionally, we examined whether the results of the whole sample (including four hospitals) were comparable to those of a subsample including only physicians from the two academic hospitals that offered the coaching intervention (ie, EMC and LUMC). We conducted repeated measures analyses for each outcome variable. Coaching attitude, that is, the degree to which one believes coaching is beneficial or helpful, was included in the analyses as control variable. The results of the analyses with the subsample showed some small differences with those of the analyses with the whole sample. Here, we find slightly stronger effects for supervisor support (ie, decrease in control group), cynicism (ie, significant increase in control group while only marginally significant result in original analyses) and work engagement (ie, increase in coaching group), all in the same direction of the results including the complete sample as shown by post hoc comparisons.

Department affiliation; We compared physicians affiliated with the paediatrics department with physicians affiliated with other departments on contextual variables that could potentially influence treatment effectiveness, that is, experienced competition and psychological safety. Experienced competition referred to the amount of competition experienced between coworkers and was measured with five items from Van Vianen.[39] Psychological safety referred to 'a shared belief held by members of a team that the team is safe for interpersonal risk taking'[32] allowing team members to express ideas, concerns or errors and was measured with nine adapted items from Edmondson[40] and van Dyck.[41] We conducted ANOVA with competition and psychological safety measured at baseline as outcome variables. The analyses revealed that our two groups, paediatricians (n=89) vs 'other' (n=25) did not differ with regard to both competition and psychological safety, with both p>0.05.

### Conclusion results

These analyses revealed that participants in the coaching group experienced gains, including decreased job demands, increased personal resources and a reduction of burn-out symptoms: participants perceived less job insecurity and work-family conflict, reported more autonomy and stronger personal resources, and showed a decrease in exhaustion, which is the main component of the burn-out syndrome. The additional analyses conducted to test for potential effects of hospital or department affiliation on the intervention effectiveness indicated no drastic changes compared with the original analyses except that—when controlling for hospital affiliation—participants in the coaching group reported increased

**Table 3** Correlations between the study variables for the intervention and control group at baseline (T1) in a study on coaching effectiveness for medical residents and specialists, 2017–2018†

| Study Variables | 1 | 2 | 3 | 4 | 5 | 6 | 7 | 8 | 9 | 10 | 11 | 12 | 13 |
|---|---|---|---|---|---|---|---|---|---|---|---|---|---|
| 1.Coaching attitude | 1 | 0.284* | -0.032 | 0.015 | -0.024 | -0.088 | -0.094 | -0.218 | -0.203 | 0.172 | -0.014 | 0.006 | 0.092 |
| 2.Workload | -0.050 | 1 | -0.089 | 0.322* | 0.015 | -0.124 | -0.209 | -0.017 | -0.067 | 0.120 | 0.299* | 0.168 | -0.066 |
| 3.Job insecurity | 0.128 | 0.191 | 1 | -0.066 | -0.458** | -0.278* | 0.095 | -0.511** | -0.405** | -0.220 | 0.225 | 0.221 | -0.285* |
| 4.Work-family conflict | -0.142 | 0.454** | -0.071 | 1 | 0.038 | -0.045 | 0.131 | 0.060 | -0.107 | -0.004 | 0.341** | 0.207 | -0.100 |
| 5.Autonomy | -0.025 | -0.333* | -0.335* | -0.232 | 1 | 0.161 | -0.036 | 0.382** | 0.224 | 0.142 | -0.177 | -0.141 | 0.290* |
| 6.Colleague support | 0.234 | -0.163 | -0.153 | -0.040 | 0.348** | 1 | -0.100 | 0.418** | 0.348** | 0.137 | -0.463** | -0.510** | 0.557** |
| 7.Supervisor support | -0.026 | 0.020 | -0.379** | -0.035 | 0.341** | 0.233 | 1 | 0.154 | -0.031 | -0.076 | 0.140 | -0.032 | 0.027 |
| 8.PsyCap | -0.173 | -0.370** | -0.463** | -0.199 | 0.486** | 0.325* | 0.401** | 1 | 0.607** | 0.242 | -0.365** | -0.576** | 0.627** |
| 9.Self-compassion | -0.176 | -0.276* | -0.422** | -0.260 | 0.244 | 0.362** | 0.427** | 0.512** | 1 | 0.086 | -0.545** | -0.397** | 0.388** |
| 10.Psych. flexibility | -0.238 | 0.031 | -0.170 | 0.189 | 0.415** | 0.256 | 0.212 | 0.273* | 0.209 | 1 | -0.094 | -0.187 | 0.407** |
| 11.Exhaustion | 0.064 | 0.493** | 0.313* | 0.411** | -0.299* | -0.222 | -0.318* | -0.363** | -0.439** | -0.157 | 1 | 0.602** | -0.570** |
| 12.Cynicism | -0.111 | 0.355** | 0.356** | 0.005 | -0.339** | -0.328* | -0.373** | -0.473** | -0.286* | -0.337* | 0.686** | 1 | -0.617** |
| 13.Work engagement | 0.228 | -0.164 | -0.228 | -0.041 | 0.309* | 0.349** | 0.491** | 0.451** | 0.303* | 0.403** | -0.482** | -0.712** | 1 |

Above the diagonal: coaching group; below the diagonal: control group.
†The following significance values are used: *p<0.05; **p<0.01.
PsyCap, psychological capital; psych. flexibility, psychological flexibility.

**Table 4** Summary of results for repeated measures analyses and preintervention and postintervention means for the intervention group in a study on coaching effectiveness for medical residents and specialists, 2017–2018

| Time x group interaction for study variables | Mean square | F | P value | $\eta_p^2$ | Preintervention mean (SD) | Postintervention mean (SD) | Df | t | P value |
|---|---|---|---|---|---|---|---|---|---|
| Job demands | | | | | | | | | |
| Workload | 0.21 | 0.84 | 0.362 | 0.007 | 3.48 (0.67) | 3.31 (0.61) | 56 | 1.97 | 0.053 |
| Job insecurity** | 6.07 | 10.99 | 0.001 | 0.090 | 4.24 (1.33) | 3.61 (1.46) | 56 | 4.10 | 0.000 |
| Work–family conflict** | 4.60 | 8.33 | 0.005 | 0.070 | 4.85 (1.05) | 4.34 (1.12) | 56 | 4.36 | 0.000 |
| Job resources | | | | | | | | | |
| Autonomy** | 3.41 | 7.56 | 0.007 | 0.064 | 4.39 (1.03) | 4.89 (1.06) | 56 | −4.19 | 0.000 |
| Colleague support* | 1.68 | 4.68 | 0.033 | 0.040 | 5.33 (.96) | 5.56 (0.79) | 56 | −1.94 | 0.057 |
| Supervisor support* | 3.79 | 5.60 | 0.020 | 0.048 | 4.63 (1.51) | 4.82 (1.35) | 56 | −1.28 | 0.207 |
| Personal resources | | | | | | | | | |
| PsyCap** | 2.57 | 17.92 | 0.000 | 0.139 | 4.83 (0.69) | 5.16 (0.65) | 56 | −4.08 | 0.000 |
| Self-compassion** | 1.26 | 10.00 | 0.002 | 0.083 | 3.07 (0.60) | 3.27 (0.52) | 56 | −2.72 | 0.009 |
| Psych. flexibility | 0.34 | 1.80 | 0.182 | 0.016 | 3.43 (0.63) | 3.47 (0.65) | 56 | −0.53 | 0.600 |
| Outcomes | | | | | | | | | |
| Exhaustion** | 6.20 | 15.94 | 0.000 | 0.126 | 2.75 (1.08) | 2.25 (0.79) | 56 | 4.00 | 0.000 |
| Cynicism* | 2.53 | 5.44 | 0.022 | 0.047 | 2.11 (1.08) | 1.90 (0.75) | 56 | 1.46 | 0.151 |
| Work engagement† | 0.69 | 3.19 | 0.077 | 0.028 | 5.08 (0.78) | 5.28 (0.59) | 56 | −2.19 | 0.033 |

The following significance values are used: †p<0.10; *p<0.05; **p<0.01.

$\eta_p^2$ refers to the degree to which variability among observations can be attributed to conditions controlling for the subjects' effect that is unaccounted for by the model.

Df for the time x group interaction=1 for all study variables and 111 for the error(time).

work engagement while no such change occurred in the control group. For all other outcome variables, neither hospital nor department affiliation influenced the effect of the intervention in a significant way, allowing us to conclude that the effect of the intervention is largely stable across the hospital organisations and department affiliations involved in this study.

## DISCUSSION
### Principal findings
Burn-out rates among medical residents and specialists are on the rise.[2] Consequently, calls for action that target the professional culture and the working environment (eg, excessive job demands) in the medical profession have been put forward.[42–45] While urgently needed, system-level changes take time. Therefore, it is imperative to develop effective measures that boost resources in order to empower physicians to effectively deal with the extreme demands they face. Although coaching is frequently advised as an intervention for physicians with burn-out, surprisingly, research on its effectiveness to create personal resources and prevent burn-out in the medical field barely exists.[10 46–48] Potential remedies for physician burn-out that have been put forward tend to be programmes that focus on curing the symptoms of burn-out, rather than preventing its onset. That is, these programmes focus on mindfulness, resilience or coping.[49–51] Here, we have shown that individual coaching is a promising route to both resolve and prevent burn-out symptoms from residency onwards. In other words, coaching can kill two birds with one stone. Physicians in the coaching group reported a decrease in exhaustion, the primary symptom and starting point of burn-out.[52] Additionally, physicians showed increases in the personal resources psychological capital and self-compassion, both strong predictors of employee well-being and performance.[53–55] In line with the JD-R model,[19] we may conclude that equipping physicians with personal resources can be a decisive factor in the prevention of burn-out. That is, when physicians expand their personal resources, their ability to impact the environment increases,[18] enhancing the chance that they will feel equipped to face stressful job demands and ultimately preventing burn-out.

### Strengths and weaknesses
To our knowledge, our study provides first evidence from a controlled intervention study on the effectiveness of coaching in both medical residents and specialists. Additionally, the two-wave design including a control group together with the additional analyses we conducted allow for a sound interpretation of the intervention effects demonstrating meaningful changes in a group of physicians (in training) who are motivated to accept assistance. However, it should be noted that the current study is limited by its quasi-experimental design.

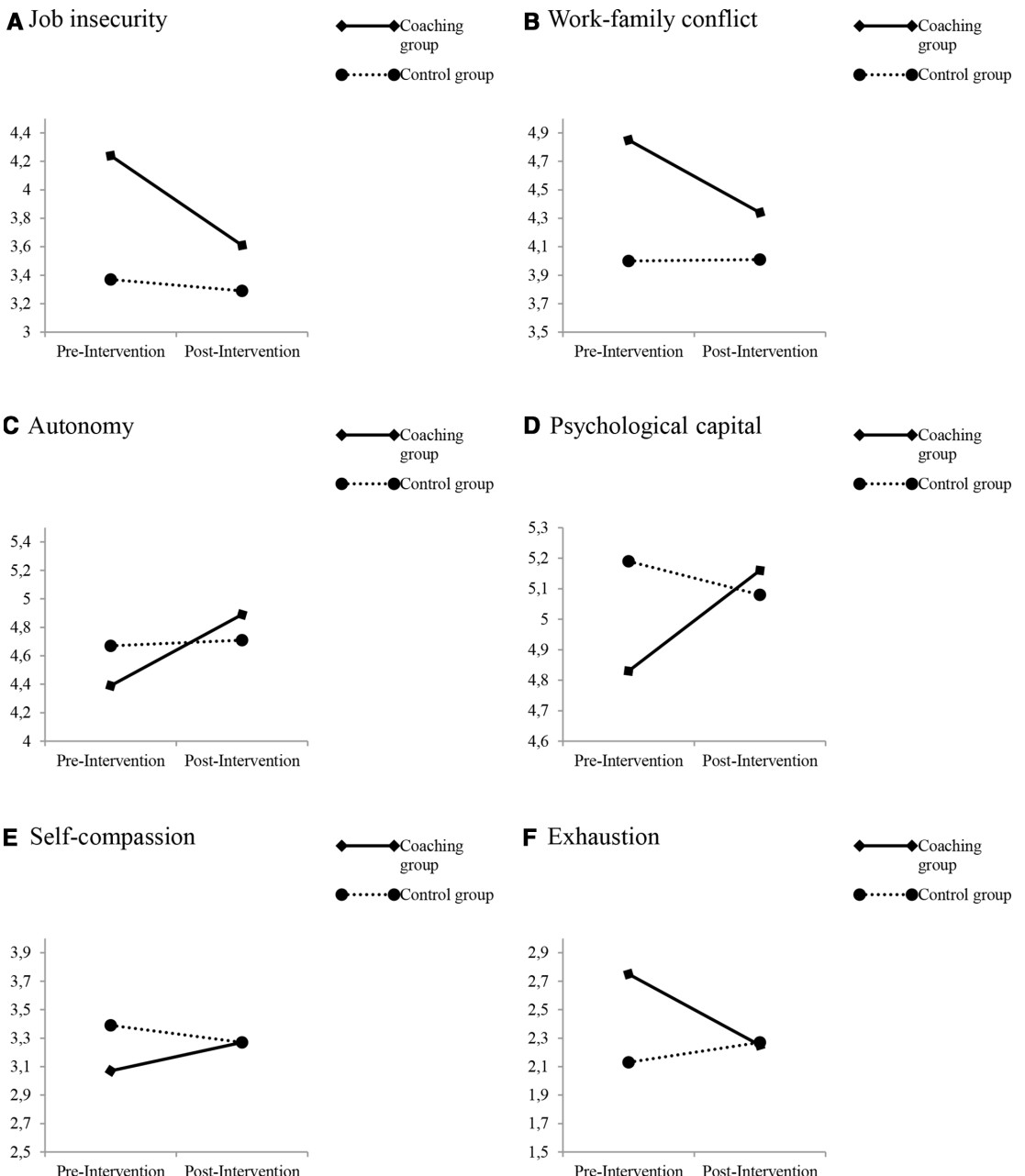

**Figure 2** Graphic representation of the outcomes at baseline and follow-up measurement for the coaching group and the control group in a study on coaching effectiveness for medical residents and specialists.

The initial differences between the groups may be the result of appropriate self-selection or may point towards a regression to the mean effect. As such, the implications of our study should be read with care. Second, although our analyses did not suggest that hospital or department affiliation influenced treatment effectiveness greatly, the multisite character of the study including different hospital and department affiliations in the groups limits our study's potential to draw causal conclusion. Third, our study design does not allow long-term inferences of coaching effectiveness. And finally, the coaching group consisted exclusively of paediatric residents and physicians. Consequently, more research is needed that evaluates the effectiveness of coaching in different specialties,

allowing broader generalisation for coaching effectiveness among healthcare professionals.

### Strengths and weaknesses in relation to other studies

Intervention studies in healthcare are scarce. However, a recent study investigating the effects of coaching on physician well-being and distress has found that specialists that received 3.5 hours of coaching by telephone showed a reduction in burn-out symptoms and improvements in overall quality of life and resilience.[10] While this study highlights the potential of coaching for specialists, the coaching method is not comparable to face-to-face coaching which makes comparison to our study difficult. Both studies, however, show that coaching leads to a

reduction in burn-out symptoms. Importantly, our study adds evidence that coaching improves well-being and fosters personal resources among residents too. These results suggest that coaching can be beneficial to healthcare professionals from residency onwards.

## Possible explanations and implications

Our study provides initial evidence that coaching may also function as a preventive tool through development of personal resources rather than a cure only. It also shows that only six individual coaching sessions, can reduce burn-out symptoms. We, therefore, hope that our results inspire healthcare practitioners and policy-makers to prioritise prevention rather than symptom alleviation. While collective action is sorely needed to bring changes on a system level, interventions like coaching can empower the whole spectrum of healthcare professionals from residents onwards to impact the healthcare system and eventually improve quality of care.

## Unanswered questions and future research

This study shows that professional coaching can reduce burn-out symptoms and strengthen personal resources. However, it is unclear how robust these effects are over time, and if effects can be generalised across different medical specialties. Additionally, the working mechanisms of coaching are yet to be discovered, making these important inquiries for the future.

**Author affiliations**
[1]Work and Organizational Psychology, University of Amsterdam, Amsterdam, The Netherlands
[2]Pediatrics, Erasmus Medical Center - Sophia Children's Hospital, Rotterdam, The Netherlands
[3]School of Business and Economics, Vrije Universiteit Amsterdam, Amsterdam, The Netherlands
[4]Pediatric Hematology, Erasmus Medical Center - Sophia Children's Hospital, Rotterdam, The Netherlands
[5]Pediatrics, Leiden University Medical Center, Leiden, The Netherlands
[6]Pediatrics/Pediatric Intensive Care Unit, Erasmus Medical Center - Sophia Children's Hospital, Rotterdam, The Netherlands

**Acknowledgements** The authors wish to thank all the medical residents and attending physicians as well as the coaches who participated in this study. The authors also want to thank Axel Willemse, Sarah van den Hee, Dr Machteld van den Heuvel and Dr Angelique Bakker-Pieper for their contribution to the design and implementation of the study.

**Collaborators** The Challenge & Support Research Network: AMC van Rossum, WJW Kollen, RGM Bredius, AJ Heesterman, MA van Houten, MJE Walenkamp, AAM Zandbergen, SCE Schuit, J EC Bromberg, A Willemse, SM van den Hee, M van den Heuvel, and A Bakker-Pieper.

**Contributors** All authors made substantial contributions to the conception and design, and the collection and interpretation of the data. LS, AvV and JK analysed the data and interpreted the data together with TT, APJdP and MDH. All authors reviewed and approved the final manuscript.

**Funding** The authors have not declared a specific grant for this research from any funding agency in the public, commercial or not-for-profit sectors.

**Competing interests** None declared.

**Patient and public involvement** Patients and/or the public were not involved in the design, or conduct, or reporting, or dissemination plans of this research.

**Patient consent for publication** Not required.

**Ethics approval** The institutional Ethics Review Board of the University of Amsterdam gave ethical approval for this study, on 12 December 2016; document 2016-WOP-7521.

**Provenance and peer review** Not commissioned; externally peer reviewed.

**Data availability statement** Data are available on reasonable request. Data and statistical code are available on request. Any queries should be directed to the first author.

**ORCID iD**
Lara Solms http://orcid.org/0000-0002-1080-3064

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
