## [Reviewer comments · BMJ Open]

ARTICLE DETAILS

TITLE (PROVISIONAL)	Turning the tide: a quasi-experimental study on a coaching intervention to reduce burnout symptoms and foster personal resources among medical residents and specialists in the Netherlands
AUTHORS	Solms, Lara; van Vianen, Annelies; Koen, Jessie; Theeboom, Tim; de Pagter, Anne; De Hoog, Matthijs

VERSION 1 – REVIEW

REVIEWER	Cheryl Regehr University of Toronto Canada
REVIEW RETURNED	23-Aug-2020

GENERAL COMMENTS	This paper presents the results of an intervention study aimed at reducing burnout among physicians. Given the increased concerns about physician wellness and evidence from across the world that physicians are experiencing significant levels of distress related to job demands, this is a critical area of inquiry and the results of the study are promising indeed. Overall the paper is well written and the study appropriately conducted given the challenges and demands of real world research. Scales selected are appropriate, as are statistical methods. I have a few minor suggestions for improvement. • The abstract indicates that the coaching occurred over a period of approximately 10 months – greater information about the average length of time between coaching sessions and the nature of the coaching would be useful to allow others to replicate the study or implement a similar intervention.• The sentence in the abstract “The coaching group (68%....)” needs further clarification – as it stands it appears to be inconsistent.• As the authors indicate, the control group has some challenges. The authors have applied various statistical methods to address these challenges. Nevertheless, the pre-post group comparisons in the study group alone provides useful information in demonstrating changes in a highly stressed group of individuals that is motivated to accept assistance, this could be further emphasized.• The authors identify that the design did not allow for the measurement of the results over time. Although, given the time elapsed since the intervention, this would seem to be relatively simple to address by contacting the participants and requesting
--

	that survey instruments be completed at T3. These results could be presented in a follow-up brief report in the journal.
--	--

REVIEWER	Tyra Fainstad University of Colorado School of Medicine United States of America
REVIEW RETURNED	12-Oct-2020

GENERAL COMMENTS	The biggest revision to make before acceptance is to flesh out the methods section about the actual coaching sessions. Where did the coaches come from? How/why did you choose them? What was their training? How were the coaching sessions performed (in person, if so where? telephone? online? at home? At work?)? How long was each session? How far apart on average were they? Answering these questions will allow for the study findings to be replicated and used in other institutions. The rest of my revisions are minor:  -Wording of the second paragraph of the intro is a bit awkward. -page 7 line 30 would not use phrase "physicians needs first" which implies over others needs or even pt care (might just say prioritize physicians). -Please define coaching in general before your specifics on page 7, line 42. A professional or life coaching definition and how it's different than other mental health resources is missing here. -Please define "personal resources" in the intro before you refer to them on page 8 line 12. I was not familiar with this term and it was confusing. Resources in what? -page 12 lines 31-36: were these attitudes towards coaching assessed at baseline (i.e. before coaching?)? Or after (theoretically, participants attitudes towards coaching would change after they had some in this study). -Page 12 line 44-46: not true. Randomization would have been feasible, you could have randomized your volunteers (and then staggered your coaching intervention and just surveyed them in the middle) - don't have to state this, but wouldn't say "not feasible". -Page 21 line 10: Don't say consequently again. You just said it. Page 21 lines 19-24 - is this true? "mostly"? might get rid of that adjective, hard to prove. Page 21 line 31 - change word condition to "group" or "inter
---

VERSION 1 – AUTHOR RESPONSE

Reviewers' comments

Reviewer 1

This paper presents the results of an intervention study aimed at reducing burnout among physicians. Given the increased concerns about physician wellness and evidence from across the world that physicians are experiencing significant levels of distress related to job demands, this is a critical area of inquiry and the results of the study are promising indeed.

Overall the paper is well written and the study appropriately conducted given the challenges and demands of real world research. Scales selected are appropriate, as are statistical methods.

Response

We thank the reviewer for this comment.

The abstract indicates that the coaching occurred over a period of approximately 10 months – greater information about the average length of time between coaching sessions and the nature of the coaching would be useful to allow others to replicate the study or implement a similar intervention.

Response

We thank the reviewer for this comment. In this real-life study, all visit intervals in the individual coaching trajectories were determined by the client and data on these – apart from the first and the last session – were not available to the researchers. However, coaching trajectories were completed within approximately 10 months ($M = 7.98$, $SD = 2.81$; calculation based on 30.44 days/month) with relatively few clients who needed more time to complete their coaching trajectory. Table 1 below shows the length of coaching trajectories and the number of participants who completed their coaching within a specific timeframe. Further, we included the following description in the methods section to give a more thorough description of the duration of the coaching trajectories and to clarify that the time in between coaching sessions varied between participants and was subject to the client (page 9, lines 18-24): “Constraints were set only with regard to the overall outline of the coaching program. That is, coaching was set to a maximum of 6 (1 or 1.5 hour long) sessions and coaches and participants were encouraged to complete the coaching trajectories within a period of approximately 10 months but could stretch their trajectories if necessary ($M = 7.98$, $SD = 2.81$), which only few participants did. All participants started their coaching trajectory individually depending on the availability of their coach. Time in between coaching sessions was determined by the participants – and hence varied – and was further not registered.”

Table 1. Length of coaching trajectories and number of participants

Length of trajectories	Number of participants
< 12 weeks	4
12 - 24 weeks	8
25 - 36 weeks	14
37 - 48 weeks	27
> 48 weeks	4

With regard to the nature of the coaching, clients and coaches were free to pick the topics discussed based on the client’s coaching query, the coaching methods applied, and the speed of the trajectories. On purpose, there was great freedom for both the client and the coach in shaping the coaching. Because an important premise of successful coaching is that the coach and the client agree on the goals to achieve, as well as the means to achieve them, - we largely avoided regulations to the coaching process (such as the topics of the coaching, the coaching method or the speed of the trajectories) that might have stood in the way of such consensus. Also, in line with the recommendations made by reviewer 2, we revised the method section to provide more information on the nature of the coaching as well as the background of the professional coaches.

To provide more information on the nature of the coaching itself we added a paragraph titled “The coaching process” containing the following information, some of which was mentioned earlier (page 9,

lines 12-25): “Coaches and participants received ample freedom to shape the coaching program according to coaches’ professional methods and participants’ needs. Because an important premise of successful coaching is that the coach and the client agree on the goals to achieve, as well as the means to achieve them, 2-3 we largely avoided regulations to the coaching process (such as the topics of the coaching, the coaching method or the speed of the trajectories) that might have stood in the way of such consensus. Constraints were set only with regard to the overall outline of the coaching program. That is, coaching was set to a maximum of 6 sessions and coaches and participants were encouraged to complete the coaching trajectories within a period of approximately 10 months but could stretch their trajectories if necessary ($M = 7.98$, $SD = 2.81$), which only few participants did. All participants started their coaching trajectory individually depending on the availability of their coach. Time in between coaching sessions was determined by the participants – and hence varied – and was further not registered. All coaching sessions took place face-to-face and outside of work at the coach’s workspace.”

To provide more information on the background and recruitment of the coaches we added the following information after having stated that coaches were selected based on a number of relevant criteria (page 9, lines 3-8): “Specifically, all coaches were selected based on their senior level of coaching experience, their experience with physician-clients, positive references from previous physician clients, and accredited coaching training. The selection committee consisted of a coaching professional, a senior human resources manager, and the medical specialist and initiator of the coaching program.”

The sentence in the abstract “The coaching group (68%....)” needs further clarification – as it stands it appears to be inconsistent.

Response

We agree with the careful comment of the reviewer and revised this sentence in the abstract. The ratio in brackets now clearly refers to the response rate in both the control and the coaching group and precludes misinterpretation (page 4, line 2-5): “The coaching group (response rate 68%, 57 physicians, 10 men, 47 women) reported a reduction in burnout symptoms and an increase in personal resources after the coaching intervention, while no such changes occurred in the control group (response rate 35 %, 15 men, 42 women), as indicated by significant Time x Group interactions, all p ’s $< .01$.”

As the authors indicate, the control group has some challenges. The authors have applied various statistical methods to address these challenges. Nevertheless, the pre-post group comparisons in the study group alone provides useful information in demonstrating changes in a highly stressed group of individuals that is motivated to accept assistance, this could be further emphasized.

Response

We thank the reviewer for this valuable comment. We added information in the section strengths and weaknesses of the discussion section that emphasizes the contribution that this study can make given the challenges originating from our design (page 23, lines 2-3): “Additionally, the two-wave design including a control group together with the additional analyses we conducted allow for a sound interpretation of the intervention effects demonstrating meaningful changes in a group of physicians (in training) who are motivated to accept assistance.”

The authors identify that the design did not allow for the measurement of the results over time. Although, given the time elapsed since the intervention, this would seem to be relatively simple to address by contacting the participants and requesting that survey instruments be completed at T3. These results could be presented in a follow-up brief report in the journal.

Response

We agree with the reviewer's comment that a follow-up survey at this point would provide valuable data to measure the long-term effects of the coaching intervention. The relatively long time (>48 months) that has passed since the completion of the coaching program however poses practical and methodological challenges.

First, it is likely that the response rate of the follow-up would be small due to a drop in engagement and practical reasons (e.g., change of employer and email-address of participants). Second, because of the relatively long time that has passed since our last measurement, it would be necessary to control statistically for a number of potential changes in the clients' professional and private lives that might influence our outcome variables. This will inevitably lead to a decreased power for conducting our analyses.

We therefore believe that a follow-up measurement at this point is not feasible with the current study sample due to both methodological and practical constraints.

Again, we agree that a follow-up measurement would lead to valuable information on the long-term effects of coaching. We are currently planning a new coaching intervention study among a larger sample of medical residents and specialists from various specialties where we intend to measure the long-term effects of coaching (i.e., six months after completion of coaching).

Reviewer 2

The biggest revision to make before acceptance is to flesh out the methods section about the actual coaching sessions. Where did the coaches come from? How/why did you choose them? What was their training? How were the coaching sessions performed (in person, if so where? telephone? online? at home? At work?)? How long was each session? How far apart on average were they? Answering these questions will allow for the study findings to be replicated and used in other institutions.

Response

We thank the reviewer for her time reviewing our manuscript. We have revised the methods section so that it now consists of a more in-depth description of the coaches, the recruitment process, and the nature of the coaching.

To provide more information on the background and recruitment of the coaches we added the following information after having stated that coaches were selected based on a number of relevant criteria (page 9, lines 3-8): "Specifically, all coaches were selected based on their senior level of coaching experience, their experience with physician-clients, positive references from previous physician clients, and accredited coaching training. The selection committee consisted of a coaching professional, a senior human resources manager, and the medical specialist and initiator of the coaching program."

To provide more information on the nature of the coaching itself we added a paragraph titled "The coaching process" containing the following information (page 9, line 12-25): "Coaches and participants received ample freedom to shape the coaching program according to coaches' professional methods and participants' needs. Because an important premise of successful coaching is that the coach and the client agree on the goals to achieve, as well as the means to achieve them, 2-3 we largely avoided regulations to the coaching process (such as the topics of the coaching, the coaching method or the speed of the trajectories) that might have stand in the way of such

consensus. Constraints were set only with regard to the overall outline of the coaching program. That is, coaching was set to a maximum of 6 (1 or 1.5 hour long) sessions and coaches and participants were encouraged to complete the coaching trajectories within a period of approximately 10 months but could stretch their trajectories if necessary ($M = 7.98$, $SD = 2.81$), which only few participants did. All participants started their coaching trajectory individually depending on the availability of their coach. Time in between coaching sessions was determined by the participants – and hence varied – and was further not registered. All coaching sessions took place face-to-face and outside of work at the coach's workspace.

The rest of my revisions are minor:

Wording of the second paragraph of the intro is a bit awkward.

page 7 line 30 would not use phrase "physicians needs first" which implies over others needs or even pt care (might just say prioritize physicians).

Response

We have revised the first sentence of the second paragraph of the introduction (page 6, lines 11-13): "In order to reduce the risk of physician burnout and thus warrant adequate patient care and patient safety, powerful interventions are needed that prioritize physicians' needs."

Please define coaching in general before your specifics on page 7, line 42. A professional or life coaching definition and how it's different than other mental health resources is missing here.

Response

To address the first part of this comment we have integrated a common global definition of coaching before referring to the limitations of coaching in healthcare (page 6, lines 13-17): "This is the case in professional coaching, which is commonly defined as "a result-oriented, systematic process in which the coach facilitates the enhancement of life experience and goal-attainment in the personal and/or professional life of normal, non-clinical clients." This definition of coaching acts on the assumption that coaching is a facilitative process aimed at self-directed change of the client."

Coaching as defined here encompasses both coaching to promote change at the workplace as well as coaching to promote change in a client's personal life. We use this broad definition of coaching in our study for two reasons. First, the coaching program was explicitly provided and stimulated by the workplace, which makes it likely that the starting point of the coaching was work related. Second, because there was no restriction on the topics being discussed during the coaching sessions, and because in the coaching practice it is not possible nor desirable to strictly separate the professional and personal life of a client we aimed to use a definition of coaching that was not restricted to the workplace but integrated clients' personal life.

To address the second part of this comment, we distinguish coaching from related 'helping relationships', such as mentoring and counseling (page 6, lines 17-23): "Additionally, this definition distinguishes coaching from other 'helping relationships' such as mentoring or counseling. Mentoring generally refers to a relationship between a more senior employee and a protégé aimed at offering guidance and feedback in a specific organizational context. In coaching, a coach usually does not hold a formal position within the client's organization. Additionally, our definition of coaching emphasizes a non-clinical target group, which makes it clearly distinguishable from counseling and therapy."

Please define "personal resources" in the intro before you refer to them on page 8 line 12. I was not familiar with this term and it was confusing. Resources in what?

Response

We agree with the reviewer and have added a formal definition of personal resources in the introduction (page 7, lines 11-13): "Personal resources refer to 'aspects of the self that are generally linked to resiliency and refer to individuals' sense of their ability to control and impact upon their environment successfully'"

page 12 lines 31-36: were these attitudes towards coaching assessed at baseline (i.e. before coaching)? Or after (theoretically, participants attitudes towards coaching would change after they had some in this study).

Response

We have clarified that attitudes towards coaching were assessed at baseline, before the start of the coaching intervention (page 12, lines 22-252): "We controlled for coaching attitude, i.e., the degree to which one believes coaching is beneficial or helpful, which was measured at baseline, because it can be expected that a positive attitude may contribute to the success of the intervention."

Page 12 line 44-46: not true. Randomization would have been feasible, you could have randomized your volunteers (and then staggered your coaching intervention and just surveyed them in the middle) - don't have to state this, but wouldn't say "not feasible".

Response

We thank the reviewer for the careful suggestion and agree that the word feasible might be misleading. Instead of "feasible" we now use the word "possible". We also added that this was the case due to time and financial constraints and prior agreements within the departments that all participants were intended to start their coaching around the same time (page 13, lines 2-6): "Because participation in the coaching program was voluntarily - and complete randomization of participants to conditions was not possible due to internal (i.e., financial and time) restrictions and prior agreements within the hospital organizations - we examined structural demographic differences prior to the intervention between the coaching and the control group (T1)"

Page 21 line 10: Don't say consequently again. You just said it.

Response

We thank the reviewer for this correction. We have changed the sentence (page 22, lines 4-6) into: "Therefore, it is imperative to develop effective measures that boost resources in order to empower physicians to effectively deal with the extreme demands they face."

Page 21 lines 19-24 - is this true? "mostly"? might get rid of that adjective, hard to prove.

Response

We agree with this comment. We adapted the wording accordingly (page 22, lines 8-10): "Potential remedies for physician burnout that have been put forward tend to be programs that focus on curing the symptoms of burnout, rather than preventing its onset."

Page 21 line 31 - change word condition to "group" or "inter"

Response

We agree with the reviewer's suggestion and changed the word "condition" into "group" (page 22, lines 13-15): "Physicians in the coaching group reported a decrease in exhaustion, the primary symptom and starting point of burnout."

Cook TD, Campbell DT. The design and conduct of true experiments and quasi-experiments in field settings. In Mowday RT, Steers RM, eds. Reproduced in part in *Research in Organizations: Issues and Controversies*. Santa Monica, CA: Goodyear Publishing Company; 1979.

McKenna DD, Davis SL. Hidden in plain sight: The active ingredients of executive coaching. *Ind Organ Psychol*. 2009;2:244-260.

Bordin ES. The generalizability of the psychoanalytic concept of the working alliance. *Psychother - Theor Res*. 1979;16:252-260.

Grant AM. The impact of life coaching on goal attainment, metacognition and mental health. *Soc Behav Personal*. 2003;31:253-263.

Theeboom T, Van Vianen AE, & Beersma B. A temporal map of coaching. *Front Psychol*. 2017;8:1352.

Theeboom T. *Workplace coaching: Processes and effects*. [PhD thesis]. Amsterdam, The Netherlands: University of Amsterdam;2016.

Noe RA. An investigation of the determinants of successful assigned mentoring relationships. *Pers Psychol*. 1988;41:457-479.

Hobfoll SE, Johnson RJ, Ennis N, Jackson AP. Resource loss, resource gain, and emotional outcomes among inner city women. *J Pers Soc Psychol*. 2003;84:632-643.